# Investigation of Color and Bioactive Compounds of Different Colors from Pansy (*Viola × wittrockiana* Gams.) Dried in Hot Air Dryer

**Deniz Hazar** [1,*] **, Ismail Boyar** [2] **, Cuneyt Dincer** [3,4] **and Can Ertekin** [2]

1 Department of Horticulture, Faculty of Agriculture, Akdeniz University, Antalya 07070, Turkey
2 Department of Agricultural Machinery and Technologies Engineering, Faculty of Agriculture, Akdeniz University, Antalya 07070, Turkey
3 Department of Food Processing, Akdeniz University Finike Vocational School, Antalya 07740, Turkey
4 Food Safety and Agricultural Research Center, Akdeniz University, Antalya 07070, Turkey
* Correspondence: dhazar@akdeniz.edu.tr

**Abstract:** The popularity of edible flowers is increasing day by day and new solutions are sought due to their short shelf life. For this purpose, in this study, four different colors of *Viola × wittrockiana Gams.*; white (Fino Clear White), orange (Delta Pure Orange), bordeaux/mauve rose (Mammoth Rocky Rose), and yellow (Delta Premium Pure Lemon) flowers were dried at drying air temperature of 60, 70, and 80 °C with a convective hot air dryer. Color (L*, a*, b*, C*, h°, ΔE, and BI), drying time and bioactive compounds (Total Phenolic Content (TPC), anthocyanin content (AC), and antioxidant activity (AO) values were measured). The orange flowers showed the fastest drying (78 min at 80 °C). The lowest total color change (ΔE) (4.58 at 70 °C) and browning index (BI) (9.58 at 60 °C) values were observed in all drying processes of white flowers. The highest AC was determined in bordeaux flowers in both fresh (2.4 mg malvidin glucoside/g) and dried (25.57 mg malvidin glucoside/g at 60 °C) samples. The AO decreased in all samples depending on the temperature increase, it was found that the most beneficial result in terms of bioactive compounds was the bordeaux flowers dried at drying air temperature of 60 °C.

**Keywords:** edible flower; convective drying; *Viola × wittrockiana*; pansy; bioactive compounds; antioxidant activity

## 1. Introduction

The sources of edible flowers can be considered as flowers of fruits, vegetables, medicinal plants, and ornamental plants [1]. It has been reported that the number of edible flowers varies by region, however, there are 97 families, 100 genera, and 180 species worldwide [2]. Recently, some ornamental plants, whose flowers are edible, have become popular as edible flowers as well as due to their extensive use as ornamental plants. The pleasant feelings evoked by flowers in a bouquet, pot, or park can also be felt with an attractive appearance in plates, salads, meals, and drinks decorated with edible flowers [3]. Begonia (*Begonia boliviensis*), borage (*Borago officinalis*), carnation (*Dianthus caryopyllus*), centaurea (*Centaurea cyanus*), chrysanthemum (*Chrysanthemum* spp.), daylily (*Hemerocallis* spp.), French marigold (*Tagetes patula*), nasturtium (*Tropaeolum majus*), pansy (*Viola × wittrockiana*), rose (*Rosa* spp.), pot marigold (*Calendula officinalis*), and tulip (*Tulipa* spp.) are some edible ornamental flowers reported in the literature [4,5].

Edible flowers have been used to add attractive appearance, taste, color, and aroma to foods, drinks, salads, and desserts in ancient Rome, medieval France, Asia, and Central Europe, in addition to traditional medicine, and their use has continued to increase throughout history [5]. Nowadays, interest in edible flowers, which are used for the same purposes in different cultures and cuisines in the world, is increasing day by day,

especially due to cookbooks, food magazines, famous chefs, restaurants, TV programs, and websites [3,6]. Rose is the most common edible flower which is used to obtain rose water, rose oil, syrup, dessert, and jam. Dried petals are also used for flavoring in Turkish delights [7]. The richness of edible flowers in terms of nutritional content and bioactive compounds has further increased consumers' interest and attracted researchers to focus on this issue. Many studies showed that the antioxidant activity of edible flowers is mainly due to the presence of flavonoids, phenolic acids, and anthocyanins. The high antioxidant activity of edible flowers is associated with several positive effects on human health, such as anti-cancer, anti-inflammatory, neuroprotective, hepatoprotective, anti-diabetic, anti-osteoporosis, anti-hypertensive, and anti-obesity [8–11].

*Viola* × *wittrokiana* Gams., commonly known as pansy or garden pansy, is one of the most popular garden and landscape flowers due to several flower colors, flowering period from middle autumn to late spring, and cold tolerance [12–14]. The biennial species of the family Violaceae, is a hybrid resulted from the crossing of three different species: *Viola tricolor*, *Viola lutea*, and *Viola altaica*, and has a taller total plant height and larger flowers than its ancestors [13,15,16]. Pansies have edible flowers characterized by velvety texture, variety of colors, and sweet fragrant taste, and all flower parts are edible [8,17]. They can be used to garnish desserts, salads, soups, beverages, and dishes. Since these plants cannot be washed, biostimulants are used during cultivation to limit the use of fertilizers and pesticides, as well as to allow them to perform well against water deficiency [18]. The phenolic profile present in pansy includes quercetin and isorhamnetin glycosides as flavonol, apigenin glycosides as flavone, cyanidin and delphinidin glycosides as anthocyanin [19]. They can be used as functional food due to their rich antioxidant activities and bioactive component content [20]. Moreover, their consumption is associated with prevention of neurodegenerative diseases [21].

Edible flowers have a niche market, however, their market share is increasing day by day. They are sold fresh in small and hard plastic packaging [6,22]. Packaging prevents contamination and wilting and also helps to preserve their fragile structure [23]. Most of them perish rapidly due to their short shelf life (2–5 days) [24]. Numerous methods have been applied to prolong the shelf life of edible flowers after harvesting. The number of these methods has increased with time. Some of them are drying [25–31], high hydrostatic pressure application [8,32], radiation ionizing [33–36], storage at low temperatures [22,37,38], and edible nanofilms (edible coatings) [39].

Drying is one of the commonly used post-harvest methods to extend the shelf life of edible flowers [40]. This technique prevents the growth of microorganisms and enzymatic degradation, and also reduces transportation and storage costs as it reduces the weight of flowers. Some of the drying methods that are currently used in edible flowers are hot air drying, freeze drying, vacuum microwave drying, cool wind drying, sun drying, osmotic drying, and their combinations [41]. It has been reported that the color, taste, and nutrient contents are better preserved when the drying process is combined with the sulfur application [42]. This has been associated with the interaction of sulfur with different biological compounds, such as plant growth regulators, enzymes, polyamines, and nutrients, to improve heat tolerance mechanisms and to produce certain derivatives that are necessary to reduce heat stress [43]. In recent years, many studies have been conducted on promising and innovative drying technologies [44]. However, the edible flower industry mainly prefers to use the hot air-drying method [41] because it is a cheap, easy to use and control drying technique, although it adversely affects product quality [40]. Temperature and time are two important factors in terms of color preservation, flower entirety, petal shape, bioactive compounds, and sensory qualities as much as possible [44]. Marigold (*Tagetes erecta* L.), coneflower (*Echinacea purpurea* (L.) Moench), roses (*Rosa* × *hybrida* L.), carnations (*Dianthus caryophyllus* L.), daylilies (*Hemerocallis disticha* Donn.), black locust flowers (*Robinia pseudoacacia* L.) and centaurea (*Centaurea cyanus* L.) are some of the edible flowers on which drying studies have been reported [45]. However, no studies on pansies have been reported in the literature.

The aim of this study was to investigate the effects of three different drying air temperatures (60, 70, and 80 °C) in a hot air convective dryer on the bioactive compounds and color of four different pansy cultivars and in order to identify the best drying temperature which can preserve the richness of bioactive compounds and quality of pansy cultivars.

## 2. Materials and Methods

### 2.1. Sample Preparation

Four cultivars of pansies (*Viola × wittrockiana Gams.*) with different petal colors: White (Fino Clear White), orange (Delta Pure Orange), mauve rose/bordeaux (Mammoth Rocky Rose), and yellow (Delta Premium Pure Lemon), were used. The petals of all cultivars were pure color and there were no blotches on the petals. Pansy seedlings were transferred from plastic plug trays to 14 cm pots containing peat and perlite (1:1 *v/v*), where they were grown in an unheated greenhouse at Akdeniz University, Faculty of Agriculture (36°53′ N and 30°39′ E, altitude 39 m) until full bloom. Flowers were harvested three times at ten-day intervals in the spring flowering period (in 2022), between 8:00 and 10:00 in the morning. In all harvests, 300 g of high quality, stemless flowers were collected from approximately 150 different plants for each cultivar. The excess water on the flowers was removed with the help of a blotting paper and was transported to the laboratory. Undamaged, uniform, and healthy flowers were carefully selected before drying. One sample from each cultivar was taken fresh for analysis. Samples were stored in the refrigerator at 4 °C until the drying experiment. All samples (fresh and dried) were stored at −18 °C for about one week until bioactive compounds analyses.

### 2.2. Drying Method

Drying tests were carried out in a Dalle LT-27 model convective dryer with a horizontal airflow cabinet. The drying air temperature can be adjusted digitally between 30 and 90 °C. The rustproof steel device manufactured in Turkey weighs 14 kg. There are 12 trays of 390 × 283 mm in size inside the dryer with dimensions of 462 × 402 × 447 mm.

In order to know the moisture content of the products before the drying process, all samples were kept at 105 °C for 24 h in triplicate, and the moisture content was determined from the weight changes. The tests were carried out in three replications at constant drying air temperatures of 60, 70, and 80 °C and at a constant drying air velocity of 1.5 m/s horizontally applied to the trays, and the results were given as average values. The convective cabinet dryer was operated empty before reaching the set temperature value before each test. Samples were weighed by taking samples manually by digital balance (AND, GF-600) for approximately one minute every 15 min as compared to the initial conditions of the test group. Drying continued until the difference between the last weight and the previous weight was 0.2 g according to the weight measurements of all flowers.

### 2.3. Color Measurements

Color values of fresh and dried products were measured with a PCE CSM3 model color measuring hand device. At the end of the measurements, L*, a*, b*, C*, and h° values were taken from the device, and the total color change (ΔE) and browning index (BI) values were calculated. The L* represents the color's brightness, ranging from 0 to 100, while positive a* and positive b* represent red and yellow, respectively. The color saturation and h° values of 0°, 90°, 180°, and 270° in the Hunter Lab scale correspond to the colors red, yellow, green, and blue, respectively [46]. Equation (1) was used to calculate the total color difference (ΔE) between the samples. Color measurements acquired from fresh pansies were used as a reference in this equation ($L_{ref}$, $a_{ref}$, $b_{ref}$). The browning index (BI) was calculated using Equations (2) and (3) [47]:

$$\Delta E = \sqrt{\left[ \left( L^* - L_{ref} \right)^2 + \left( a^* - a_{ref} \right)^2 + \left( b^* - b_{ref} \right)^2 \right]} \qquad (1)$$

$$BI = [100(x - 0.31)] \div 0.172 \text{ where x,} \tag{2}$$

$$x = (a^* + 1.75L^*) \div (5.645L^* + a^* - 3.012b^*) \tag{3}$$

### 2.4. Bioactive Compound Analyses

2.4.1. Preparation of Extracts

Extraction of the samples was performed according to the method of [48] with some modifications. Pansies samples were crushed with a blender (Beko, BKK-2155 Maxi Hand Blender, Istanbul/Turkey), and 1 g of crushed sample was weighed in 50 mL centrifuge tube. Twenty mL of 60% ethyl alcohol (containing 0.1% HCl) was added into the tube. The tubes were placed in an ultrasonic bath (Caliskan, Ultrasonic cleaner 180 W, 40 kHz, Ankara/Turkey), and extraction was conducted at 40 kHz constant frequency, at 30 °C for 30 min. Thereafter, the samples were centrifuged in a centrifuge (Eppendorf Centrifuge 5810, Nussloch/Germany) for 15 min at $4000 \times g$ rpm, and the supernatant was obtained. Extraction was performed according to the above steps and was repeated three times. The filtrate was collected and stored at 4 °C until use.

2.4.2. Determination of Total Phenolic Content

The total phenolic content (TPC) analyses were performed using the method described by [49]. For this purpose, 0.5 mL of the extract was treated with 2.5 mL of 0.2 N Folin–Ciocalteu reagent and 2 mL of $Na_2CO_3$ (75 g/L). Then, the mixture was incubated at 50 °C for 5 min and immediately cooled. The absorbance of the final solution was recorded with a spectrophotometer (Thermo Fisher Scientific, Evaluation 160 model, UV-Vis, Madison, WI, USA) at a wavelength of 760 nm with respect to the blank solution (60% ethyl alcohol (containing 0.1% HCl)). The results were expressed as milligrams of gallic acid equivalents per gram (mg GAE/g).

2.4.3. Determination of Antioxidant Activity Using DPPH

Antioxidant activity (AO) of the samples was analyzed using the DPPH assay described by [50]. The diluted sample extract of 50 μL was added to 950 μL of diphenyl picrylhydrazil (DPPH) solution ($6 \times 10^{-5}$ M in methanol). The mixtures were shaken and kept in the dark at room temperature for 30 min. Absorbances were recorded at 515 nm (Thermo Scientific Evaluation 160 UV-Vis, USA). Trolox was used as the standard of measurement, and the antioxidant activity was reported in mg Trolox/g.

2.4.4. Total Monomeric Anthocyanin Content

Total monomeric anthocyanin content (AC) was measured using a pH-differential method described by [51] in combination with a two-buffer system that utilized potassium chloride buffer (0.025 M, pH 1.0) and sodium acetate buffer (0.4 M, pH 4.5). Extracts were diluted with buffers (pH 1.0 or 4.5) and incubated for 30 min at room temperature before absorbances were measured at 520 and 700 nm (Thermo Scientific Evaluation 160 UV-Vis, USA). Anthocyanins were quantified as milligrams of malvidin glucoside equivalents per gram (mg/g).

### 2.5. Statistical Analysis

For each temperature experiment, color measurements (9 different, total 27 measurements) and bioactive compounds (3 different, total 9 measurements) were made for each repetition of dried edible flowers in 3 repetitions. These measurements were subjected to the Duncan multiple comparison test using SPSS (Version 17; Chicago, IL, USA) statistical analysis program. The statistical differences between means were considered significant at $p < 0.05$.

## 3. Results and Discussion

### 3.1. Effects of Drying Air Temperatures on Drying Time

The initial moisture content was calculated as 86.73%, 87.72%, 87.98%, and 88.79% w.b. (wet basis) for orange, yellow, white, and bordeaux pansies, respectively. Drying continued until the final moisture content. The moisture content of white pansy flowers was 10% (w.b.) when dried at 70 and 80 °C and 25% when dried at 60 °C. Additionally, the highest moisture content (25%) was obtained at 60 °C in white flowers during the experiment (Figure 1a). At 70 and 80 °C, the moisture content of samples dropped below 10% (w.b.) in the orange pansy flowers, however, it was 15% (w.b.) at 60 °C (Figure 1b). In bordeaux pansy flowers, the moisture content at all temperatures was 10% (w.b.) (Figure 1c). In yellow pansy flowers, 10% (w.b.) moisture content was observed only in drying at 70 °C, while those dried at 60 and 80 °C had 20% moisture content (w.b.) (Figure 1d). The final moisture contents of pansies of different colors varied depending on the drying temperatures.

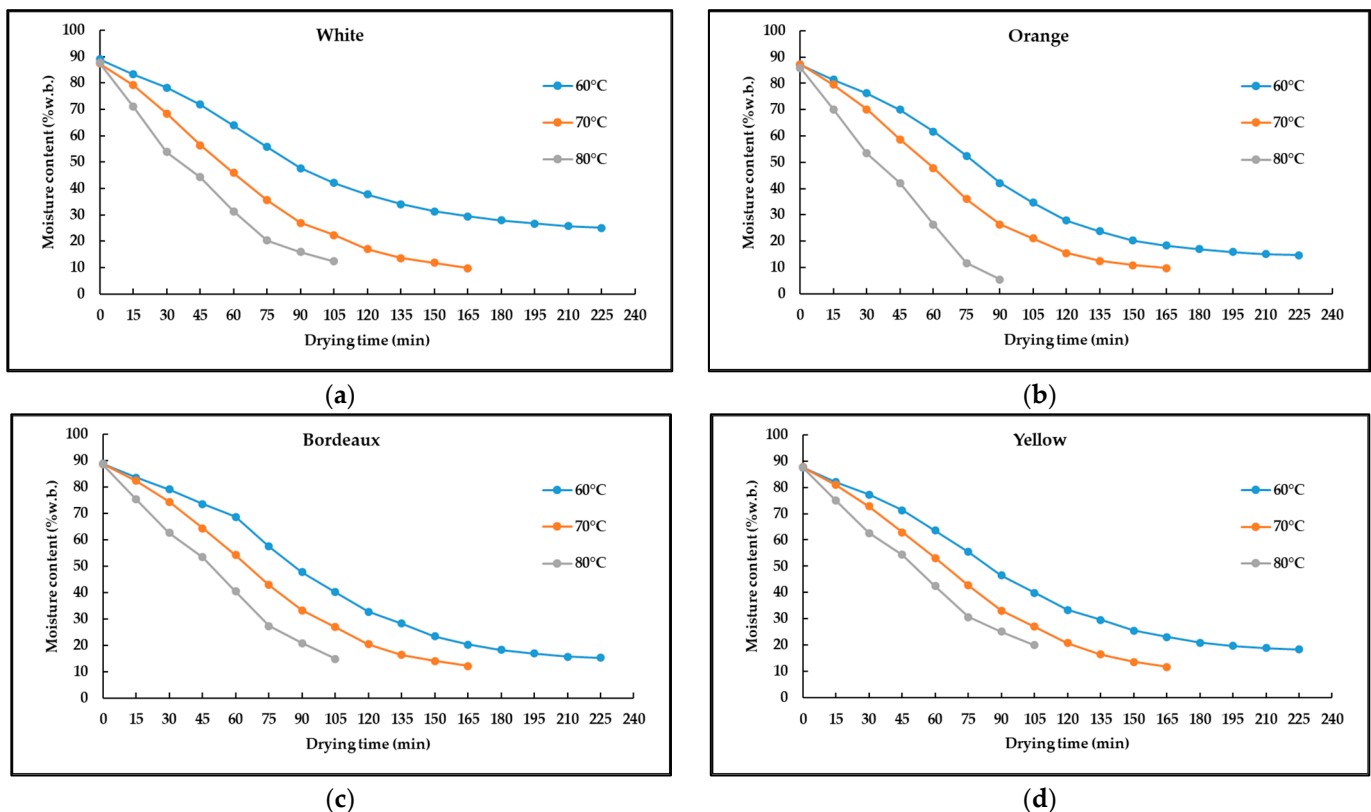

**Figure 1.** The effect of different drying air temperatures on the drying time of different colored flowers (**a**) white, (**b**) orange, (**c**) bordeaux, and (**d**) yellow pansies.

Drying is a food preservation technique that prevents microbial spoilage and slows down enzymatic activity by removing moisture from foods. Generally, foods that contain 25% moisture content or below are considered dried foods [52]. Although the safe moisture content range for dried products is accepted as 10% (w.b.) [53], it was observed that some samples reached equilibrium moisture content at higher moisture content. Bae and Lee [54] reported that 23 ± 1% moisture content was obtained in *Chrysanthemum boreale* M. flowers dried in a hot air dryer at 40, 50, and 60 °C. Our results are also in agreement with [55] using infrared drying and solar drying [56] of rose petals. The average moisture contents obtained at different drying temperatures in this study showed that bordeaux pansies had the lowest moisture content, followed by orange, yellow, and white pansies, respectively. Chen et al. [26] also observed that when two levels (1 and 2 h) of freeze drying (−35 °C, 30–50 μm Hg) and three levels (27, 37, and 47 °C) of vacuum drying were applied to two

different colored carnations and roses (red and pink), the red-colored carnation and rose had less moisture content than the pink-colored ones. According to their results, they found that higher vacuum drying temperatures resulted in lower moisture content, which agrees with our results.

The variations in moisture content of pansies of different colors as a function of drying time at different drying air temperatures showed that the moisture content of bordeaux, orange, yellow, and white pansy flowers at 60 °C drying temperature was 10, 15, 20, and 25% (w.b.), respectively. At 70 °C, drying air temperature, all of the flowers reached equilibrium moisture content by falling to 10% (w.b.) moisture content. At 80 °C, the distinctively orange flowers were the first to drop below 10% (w.b.) moisture content. It was followed by white, bordeaux, and yellow with 10, 10, and 20% (w.b.) moisture content, respectively. Drying time was calculated as 225, 165, and 105 min for all pansy colors at 60, 70, and 80 °C, respectively. Drying time decreased by 214.3% when the temperature was increased from 60 °C to 80 °C.

In our study, drying time decreased depending on the increase in drying air temperature. The edible flowers were dried by using hot air at 30, 50, 60, and 70 °C. The slowest drying occurred at 30 °C in 65 h [57]. In the infrared dryer, the rose petals were dried at three different temperatures, 50, 60, and 70 °C, and the longest drying time was 28 min at 50 °C, and the shortest was 10 min at 70 °C [55]. Consistent with our study, increasing the drying temperature (from 50 °C to 70 °C) decreased the drying time of rose petals by 280%. *Chrysanthemum boreale* M. flowers were dried with hot air at 40, 50, and 60 °C and the drying process was completed in 69 h at 40 °C and 8.5 h at 60 °C [54]. Our results were consistent with other studies carried out in hot air dryers in terms of shortening the drying time due to the increase in drying temperature, however, it had a shorter drying time than the studies conducted previously. Three different drying methods were used to dry Centaurea petals, drying with hot air at 50 °C, drying under the shade, and freeze-drying. Drying under the shade took 72 h, freeze drying took 24 h and drying with hot air was continued for 1–4 h to reach equilibrium moisture contents [45].

### 3.2. Effects of Drying Air Temperatures on Color Parameters

The effects of hot air drying on the color parameters (L*= Lightness, a* = redness, b* = yellowness, C* (chroma) = color intensity, h° = hue angle, ΔE = color change, BI = Browning index) of white, orange, bordeaux, and yellow flowers, dried flowers were determined by comparing fresh with dried flowers and are presented in Table 1.

**Table 1.** Color characteristics of pansy flowers of different colors dried at different temperatures.

| Color | Temperature, °C | L* (Lightness) | a* (Redness) | b* (Yellowness) | C* (Chroma) | h° (Hue Angle) | ΔE (Total Color Change) | BI (Browning Index) |
|---|---|---|---|---|---|---|---|---|
| White | 60 | 85.34 [c] ± 2.33 | −1.13 [a] ± 1.10 | 8.82 [b] ± 2.13 | 8.93 [b] ± 2.24 | 96.27 [b] ± 5.58 | 7.17 [b] ± 2.36 | 9.57 [a] ± 1.92 |
| | 70 | 82.32 [b] ± 2.94 | −1.14 [a] ± 0.71 | 9.41 [bc*] ± 1.72 | 9.50 [bc*] ± 1.77 | 96.47 [b] ± 3.61 | 4.58 [a] ± 2.17 | 10.73 [b] ± 1.98 |
| | 80 | 76.42 [a] ± 5.31 | −1.60 [a] ± 0.88 | 10.25 [c] ± 2.04 | 10.39 [c] ± 2.13 | 98.70 [b] ± 3.34 | 8.13 [b] ± 3.40 | 12.43 [c] ± 2.50 |
| | Fresh | 82.54 [b] ± 3.71 | 0.52 [b] ± 0.87 | 4.82 [a] ± 1.71 | 5.10 [a] ± 1.67 | 61.66 [a] ± 7.38 | - | - |
| Orange | 60 | 44.91 [b] ± 3.93 | 31.06 [b] ± 2.81 | 30.48 [a] ± 2.83 | 43.62 [a] ± 2.56 | 44.47 [a] ± 4.09 | 26.88 ± 3.78 | 151.47 ± 15.18 |
| | 70 | 44.92 [b] ± 5.08 | 30.14 [ab*] ± 3.59 | 31.27 [a] ± 5.02 | 43.50 [a] ± 5.66 | 45.88 [a] ± 3.27 | 29.05 ± 6.53 | 154.38 ± 21.65 |
| | 80 | 41.84 [a] ± 4.21 | 28.90 [a] ± 3.06 | 29.09 [a] ± 4.04 | 41.08 [a] ± 4.31 | 45.07 [a] ± 3.79 | 28.02 ± 5.21 | 154.92 ± 16.79 |
| | Fresh | 54.05 [c] ± 2.65 | 37.83 [c] ± 4.90 | 54.57 [b] ± 4.25 | 66.52 [b] ± 4.97 | 55.35 [b] ± 3.46 | - | - |
| Bordeaux | 60 | 14.63 [b] ± 2.96 | −5.47 [a] ± 3.87 | −1.73 [a] ± 2.07 | 6.08 [a] ± 3.88 | 201.80 [c] ± 26.63 | 27.77 ± 3.84 | −42.67 ± 37.89 |
| | 70 | 14.67 [b] ± 1.76 | −4.40 [a] ± 3.27 | −0.72 [b] ± 1.23 | 4.65 [a] ± 3.23 | 185.72 [b] ± 29.02 | 26.21 ± 3.26 | −28.48 ± 24.38 |
| | 80 | 12.76 [a] ± 3.35 | −3.82 [a] ± 3.27 | −1.22 [ab*] ± 1.90 | 4.23 [a] ± 3.52 | 194.60 [bc*] ± 26.37 | 27.12 ± 3.49 | −35.87 ± 46.01 |
| | Fresh | 12.38 [a] ± 2.49 | 21.99 [b] ± 5.13 | 1.98 [c] ± 1.41 | 22.11 [b] ± 5.11 | 5.30 [a] ± 5.11 | - | - |
| Yellow | 60 | 59.98 [ab*] ± 5.24 | 23.47 [c] ± 4.88 | 61.43 [ab*] ± 4.51 | 66.00 [b] ± 3.36 | 69.00 [a] ± 5.06 | 34.94 ± 6.36 | 245.01 ± 31.55 |
| | 70 | 60.83 [b] ± 4.98 | 20.08 [b] ± 4.56 | 62.73 [b] ± 4.44 | 65.97 [b] ± 4.05 | 72.18 [b] ± 4.21 | 33.15 ± 6.26 | 242.72 ± 19.98 |
| | 80 | 58.07 [a] ± 4.61 | 19.19 [b] ± 5.02 | 59.70 [a] ± 4.52 | 62.91 [a] ± 4.48 | 72.20 [b] ± 4.61 | 33.43 ± 5.80 | 241.13 ± 21.73 |
| | Fresh | 71.63 [c] ± 2.50 | 12.77 [a] ± 3.11 | 91.34 [c] ± 3.21 | 92.30 [c] ± 3.18 | 82.04 [c] ± 2.28 | - | - |

* According to the Duncan multiple range test, means with similar letters are statistically non-significant. (Results are means ± standard deviation).

Color parameters of white, orange, bordeaux, and yellow edible flowers dried at different temperatures were investigated according to the flower colors. The L* value of white flowers decreased depending on the temperature increase. There was no change in a* and h° values due to temperature changes. When grouped statistically, they were in the same group. b* and C* values increased depending on the temperature increase. The lowest ΔE and BI values were observed in the samples dried at 70 °C and 60 °C, respectively. The L* values of the fresh samples and the dried samples at 70 °C were in the same group.

The L* value was found to be higher in samples dried at 60 and 70 °C compared to 80 °C in orange-colored flowers. The a* value was lower at all drying temperatures compared to the fresh samples. The highest a* value was in 60 and 70 °C. The b* value was lower at all drying temperatures compared to fresh samples. In comparison, the C* and h° values were lower than the fresh samples at all drying temperatures. Statistically, all tested temperatures were in the same group. There was no statistically significant difference between ΔE and BI values.

The L* value in bordeaux-colored flowers decreased due to the increase in temperature, the samples dried at 60 and 70 °C were in the same group. The a*, b*, and C* values decreased compared to those recorded in fresh samples at all drying temperatures. The highest h° value 201.80° was observed in the samples dried at 60 °C. There were no statistically significant differences in ΔE and BI values. At all drying temperatures, there was an increase in L* values as compared to fresh.

The highest L* value in yellow-colored flowers was noted in the samples dried at 70 and 60 °C. The a* value showed decrease with an increase in drying temperature. The highest b* value was in 70 and 60 °C. The C* values of the samples dried at 60 and 70 °C were higher than those dried at 80 °C. On the contrary, the highest h° value was recorded in 70 and 80 °C and the lowest h° value was in 60 °C. There were no significant differences in ΔE and BI values of yellow-colored flowers.

The color of all flowers became (lightness) darker as a result of drying processes, except for the drying of white flowers at 60 °C and bordeaux flowers at 60 and 70 °C. Drying resulted in brighter (C*) colors in white flowers, with dull colors in other flowers. The h° values of white, orange, and yellow colors did not show remarkable change with drying. The colors of these flowers were close to the fresh samples. On the other hand, the

h° value in bordeaux-colored flowers was much higher (185.72–201.80°) as compared to the fresh ones (5.30°), and the color turned into a color close to blue. Figure 2 shows the visual variation in white, orange, bordeaux, and yellow pansy flowers dried in a hot air dryer (at 70 °C) compared to fresh flowers. Visually, pansy flowers dried in a hot air dryer had a darker, shriveled appearance and smaller size than fresh flowers.

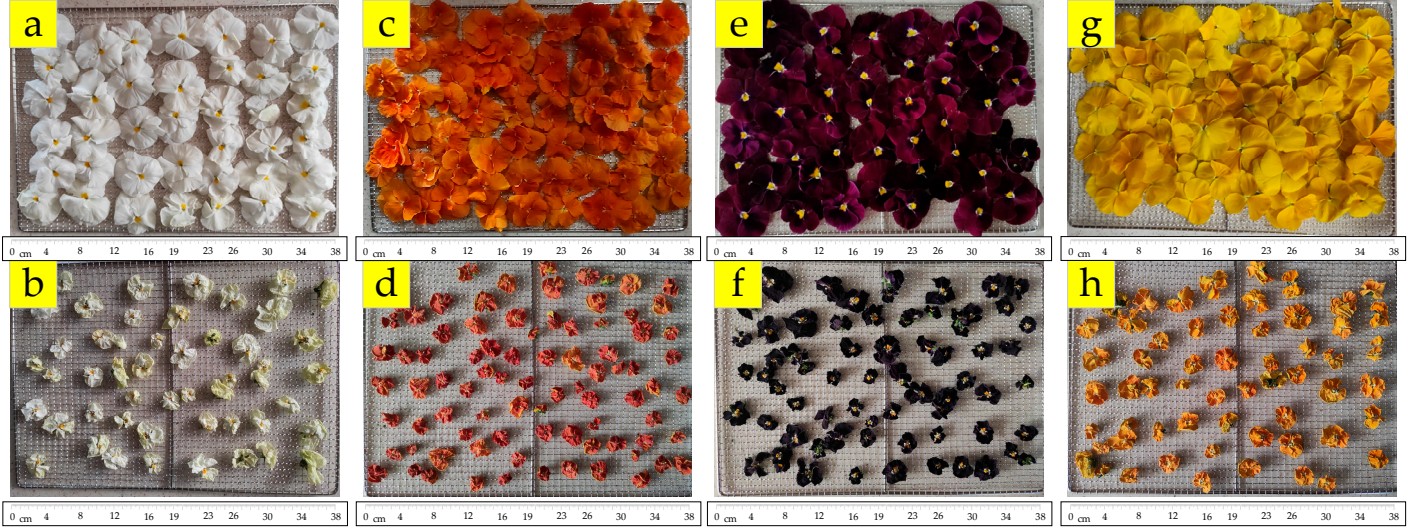

**Figure 2.** Fresh and dried (at 70 °C) white (**a,b**), orange (**c,d**), bordeaux (**e,f**), and yellow (**g,h**) samples.

Color is one of the most important factors in determining consumer preference for edible flowers, as edible flowers are used by consumers to add special color and flavor to dishes [23]. The results obtained in this study regarding bordeaux-colored flowers, which is close to red, showed similarity with Chen et al. [26], who obtained darker, bluer, and greener flowers, especially in red flowers at higher vacuum drying temperatures (application after freeze-drying) in red and pink carnations and roses.

The effects of 60, 70, and 80 °C hot air-drying temperatures on flower color parameters of pansies with different flower colors are presented in Table 1**.** The L* value decreased significantly in white flowers due to the increase in drying temperatures. There was no statistically significant difference between L* values when the temperature increased from 60 °C to 70 °C in other colors, it decreased significantly when the temperature increased to 80 °C. The a* and C* values decreased in orange, bordeaux, and yellow flowers with an increase in drying air temperatures, while they increased in white flowers. The effect of drying temperatures on b* value was directly proportional to white flowers, while there was no systematic correlation in the other flowers. The h° value increased in white and yellow flowers depending on drying temperatures, it decreased in bordeaux and orange flowers. The lowest total color change (ΔE) value was in the white pansies, with 4.58. It was followed by bordeaux- (26.21), orange- (26.28), and yellow- (33.15) colored ones, respectively. The lowest total color change (ΔE) was in 70 °C for white, bordeaux, and orange, while it was in 60 °C for orange. The lowest Browning index (BI) was in white (9.58) at 60 °C. The BI increased with an increase in temperature for white and orange flowers, while it decreased in yellow. However, BI for bordeaux was negative value (-) at all drying temperatures.

Depending on the increase in drying temperature, the L* value decreased in *Chrysanthemum boreale* dried in a hot air dryer [54], while it increased in rose petals dried in an infrared dryer [55]. In our study, similar to *C. boreale*, the L* value decreased due to the increase in temperature. In other words, the drying process caused darkening of flower color in all pansies. Another study reported that freeze-drying and hot air drying at temperatures of 50, 60, 70, and 80 °C were performed on *Begonia cucullata*, and the highest L* value was observed in freeze drying with L* value showed a decrease with the increasing

temperature in hot air drying [40]. In *C. boreale*, a* value increased with the increase in the temperature of the hot air dryer, while the b* value decreased [54]. Our results showed a contradiction with the findings obtained in these studies. A decrease was observed in all a*, b*, and C* values of dried *B. cucullata* flowers compared to fresh samples. Our results in terms of C* are consistent with the results obtained by [40], who reported a significant decrease in C* with reduction in saturation and opaque appearance at the highest drying temperature. Different drying methods were used in *Tagetes erecta*. FIR-HA (far-infrared radiation with hot air convection) application had the lowest ΔE followed by hot air drying (60 °C for 4 h) and freeze drying, respectively [30]. ΔE value of *Rose electron* was the lowest at 70 °C. In our study, the lowest total color change (ΔE) for three pansy cultivars was at 70 °C except for orange. Our findings regarding ΔE were in agreement with the outcomes of this study [55].

In general, dried flowers of all cultivars were darker, greener, and bluer than fresh ones. It also had the same effect on the increase in drying temperature. In particular, the h° angle was closer to its own colors in white, orange, and yellow, while it turned blue in bordeaux. White pansies showed the least total color change, while yellow had the most total color change. The drying temperature at which the color was best preserved (ΔE) was 70 °C. The lowest browning was found in white, with the highest browning obtained in yellow.

### 3.3. Effects of Drying Air Temperatures on Bioactive Compound Analysis

The total phenolic content, antioxidant activity, and anthocyanin content of different colors of pansies are given in Table 2. There was no significant difference in TPC white flowers according to temperature change. The lowest AO activity was in the samples dried at 70 °C, while the highest AO activity was observed at 80 °C. The highest AC was recorded in the samples dried at 70 and 80 °C.

**Table 2.** Bioactive compounds of pansy flowers in fresh and dried samples.

| Color | Temperature (°C) | Total Phenolic Content (mg/g GAE) | Antioxidant Activity (mg/g TE) | Anthocyanin Content (mg Malvidin Glucoside/g) |
|---|---|---|---|---|
| White | 60 | 14.98 [b] ± 3.04 | 4.40 [b] ± 1.27 | 0.183 [b] ± 0.04 |
| | 70 | 12.36 [b] ± 0.76 | 1.16 [a] ± 1.06 | 0.308 [c] ± 0.01 |
| | 80 | 14.11 [b] ± 0.97 | 8.31 [c] ± 1.96 | 0.300 [c] ± 0.01 |
| | Fresh | 1.98 [a] ± 0.08 | 0.29 [a] ± 0.15 | 0.001 [a] ± 0.00 |
| Orange | 60 | 74.04 [c] ± 2.66 | 240.36 [d] ± 14.85 | 0.199 [b] ± 0.02 |
| | 70 | 68.86 [b] ± 1.32 | 211.14 [c] ± 11.60 | 0.215 [b] ± 0.01 |
| | 80 | 76.89 [c] ± 1.24 | 152.78 [b] ± 16.74 | 0.226 [b] ± 0.03 |
| | Fresh | 7.37 [a] ± 0.37 | 15.88 [a] ± 1.79 | 0.001 [a] ± 0.00 |
| Bordeaux | 60 | 65.52 [b] ± 1.59 | 259.38 [c] ± 11.92 | 25.571 [c] ± 0.89 |
| | 70 | 66.65 [b] ± 2.49 | 243.22 [c] ± 12.91 | 9.474 [b] ± 0.25 |
| | 80 | 63.78 [b] ± 2.91 | 196.31 [b] ± 9.40 | 8.439 [b] ± 1.91 |
| | Fresh | 7.18 [a] ± 0.45 | 26.93 [a] ± 1.91 | 2.448 [a] ± 0.02 |
| Yellow | 60 | 61.64 [bc*] ± 0.94 | 210.39 [d] ± 7.61 | 0.196 [b] ± 0.03 |
| | 70 | 60.75 [b] ± 1.84 | 186.18 [c] ± 9.94 | 0.285 [c] ± 0.07 |
| | 80 | 63.12 [c] ± 0.86 | 153.24 [b] ± 15.43 | 0.276 [bc*] ± 0.05 |
| | Fresh | 7.44 [a] ± 0.71 | 18.61 [a] ± 4.41 | 0.001 [a] ± 0.00 |

* According to the Duncan multiple range test, means with similar letters are statistically non-significant. (Results are means ± standard deviation).

In orange pansy flowers, the lowest TPC was noted in the samples dried at 70 °C, with the highest TPC was in 60 and 80 °C. An increase in the drying temperature of orange flowers decreased the AO. The AO was 240.36, 211.14, and 152.78 mg/g TE at 60, 70, and 80 °C, respectively. The temperature change did not show significant differences in the AC of the orange flowers.

There was no significant difference in the TPC of the bordeaux-colored flowers according to the temperature change. The AO activity decreased depending on the temperature increase. Statistically, the samples dried at 60 and 70 °C were in the same group regarding AO. The highest AO activity 259.38 mg/g TE was found in the samples dried at 60 °C, while the highest AC 25.571 mg malvidin glucoside/g was recorded in the samples dried at 60 °C. Statistically, the samples dried at 70 and 80 °C were in the same group.

In yellow flowers, the highest TPC was observed in the samples dried at 80 °C with the lowest TPC was found in the samples dried at 70 °C. Similar to orange-colored flowers, there were significant differences in the AO of yellow flowers. An increase in the drying temperature had a negative effect on the AO activity, the highest AO activity 210.39 mg/g TE was found in samples dried at 60 °C, and the lowest AO activity 153.24 mg/g TE was at 80 °C. The lowest AC was found in the samples dried at 60 °C, while the highest AC was at 70 °C and 80 °C.

The experiments conducted at all drying air temperatures showed that the highest AO activity and AC were observed in bordeaux-colored flowers, while the highest TPC was noted in orange-colored flowers. The lowest TPC and AO activity were determined in the white-colored flowers at all temperatures. The lowest AC was recorded in white-, orange- and yellow-colored flower samples dried at 60 °C and in general, the anthocyanin content increased as the drying temperature increased in these pansies. The highest anthocyanin content was recorded at 60 °C in bordeaux-colored flowers. In bordeaux-colored flowers, there was an inverse negative correlation between temperature and anthocyanin content. It is seen that the effect of temperature on anthocyanins differs according to the flower type studied. As a matter of fact, it has been reported that the stability of anthocyanins is influenced by several factors, such as pH, light, co-pigmentation, sulfites, ascorbic acid, oxygen, and enzymes, as well as temperature [57]. The highest AO activity was observed in bordeaux-, orange-, and yellow-colored flowers at 60 °C, respectively. White-colored flowers had the lowest AO activity at 70 °C. At 80 °C, the highest and lowest AO activity were observed in bordeaux- (196.31 mg/g TE) and white- (8.31 mg/g TE) colored flowers, respectively, while the AO activity of orange- and yellow-colored flowers was very close to each other.

In a study on edible flowers, freeze-drying and four different drying air temperatures of 30, 50, 60, and 70 °C were used. The result of the study showed that high temperatures have negative effects on bioactive compounds except for anthocyanin. Hot air drying at 30 °C was found to be reasonable, however, it was time-consuming. The highest anthocyanin content was observed in hot air drying at 60 °C, freeze-drying, and drying at 50 °C. TPC and AO were the highest in hot air drying at 30 °C and decreased with the increasing temperature [58]. The findings of this study showed uniformity with our results in terms of AO activity. Another study was conducted on the drying of Centaurea petals by using hot air convective drying (50 °C for 1, 2, 3, 4 h), shade drying (22 °C, 3 days), and freeze drying (24 h) methods. The result obtained showed that drying in the shade gave the best values in terms of monomeric anthocyanins, flavonoids, and antioxidant activity. Hot air convective dryers produced the lowest anthocyanin and flavonoid contents [45]. A study conducted with three different colored *Viola × wittrockiana* (yellow, red, and violet) showed that violet-colored petals were found to be the most active in all solvents used for extraction and also exhibited higher total phenolic, flavonoid, and anthocyanin content compared to red and yellow ones [20].

An increase in total phenolic content was observed in rose petals dried at different temperatures compared to fresh. It is expected that the values of bioactive compounds will increase after the water is removed from dried products. The phenolic compound

degradation enzymes were inactivated due to low water activity in the dried samples, therefore, higher phenolic compounds were found in the extract [55]. Similar results were observed in this study. In another study, *Begonia cucullata* flower was freeze-dried and dried with hot air at 50, 60, 70, 80 °C. The results obtained exhibited that the highest anthocyanin contents were at 60 and 70 °C, while the highest total phenolic content was found in freeze-drying and hot air at 50, 60, 70 °C. The highest total antioxidant activity was observed in freeze-dried samples [40]. Freeze-drying and hot air drying at 30 and 50 °C were performed on red clover (*Trifolium pratense*), sweet violet (*Viola odorata*), and elderflowers (*Sambucus nigra*) flowers. While fresh samples reached the highest values in terms of total polyphenol and anthocyanin contents, the lowest values in terms of antioxidant activity (DPPH) were obtained from fresh samples. The drying application with the highest total polyphenol content was freeze-drying followed by hot air drying at 50 °C. The highest anthocyanin content was obtained from sweet violet, and there was no significant difference between drying applications. The highest antioxidant activity was obtained from drying with hot air at 50 and 30 °C, respectively. As a result of the research, the researchers reported that freeze-drying had better performance [59]. Ethanol and water were used for the analysis of bioactive compounds in freeze-dried and hot air-dried (55 °C) daylily (*Hemerocallis fulva* Linn.) samples. Ethanol being a more effective method for detecting antioxidants, freeze-drying preserved bioactive compounds better than hot air drying [60]. Another study reported that *Chrysanthemi Flos* flowers were dried with hot air at 40, 50, and 60 °C or with far infrared radiation. In the hot air-dried samples, an increase in antioxidant activity was observed with an increase in temperature. The antioxidant activity exhibited a decrease in the far infrared dried conditions. These researchers mentioned that the antioxidant activities and volatile compounds of *Chrysanthemi Flos* flowers were significantly affected by drying conditions [61].

## 4. Conclusions

Edible pansy flowers have a very short shelf life. They are generally consumed fresh. Convective hot air drying is recommended for drying of edible pansy flowers. It is a cheap and easily applicable method to preserve bioactive compounds and extend the shelf life of pansies. In this study, the optimum drying temperature, which reached equilibrium moisture content in a short time was 80 °C. Color and appearance can be improved by choosing lower drying temperatures. It was observed that the white flowers preserved these characteristics in all examined drying temperatures. While dried bordeaux-colored flowers had high anthocyanin content and antioxidant activity, orange-colored flowers had high total phenolic contents, therefore, they can be consumed as functional foods.

**Author Contributions:** Conceptualization, D.H., I.B. and C.E.; methodology, D.H., I.B. and C.E.; software, I.B. and C.E.; investigation, D.H., I.B., C.D. and C.E.; resources, D.H., C.D. and C.E.; writing—original draft preparation, D.H., I.B., C.D. and C.E.; writing—review and editing, I.B. and C.E.; visualization, C.E.; supervision, C.E. All authors have read and agreed to the published version of the manuscript.

**Funding:** This research received no external funding.

**Institutional Review Board Statement:** Not applicable.

**Informed Consent Statement:** Not applicable.

**Data Availability Statement:** Data may be available after request.

**Acknowledgments:** We would like to thank the Parks and Gardens Department of Antalya Metropolitan Municipality (white and orange pansy cultivars) and the Parks and Gardens Department of Akdeniz University (yellow and bordeaux pansy cultivars) for the pansy seedlings donation. We also thank Ibrahim BAKTIR for his careful English language corrections.

**Conflicts of Interest:** The authors declare no conflict of interest.

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
