# Peer review of "Investigation of Color and Bioactive Compounds of Different Colors from Pansy (Viola × wittrockiana Gams.) Dried in Hot Air Dryer"

_horticulturae, doi:10.3390/horticulturae9020186_

Round 1
Reviewer 1 Report
After carefully reading the manuscript, I point out some corrections below so that it is published in its best version.
the introduction should be short. The main topics must be mentioned, but in the current format it becomes very repetitive and exhausting for the reader;
please insert reference (99-100);
the formal English used in the manuscript must be improved;
please insert the drying temperature used (lines 221-223);
please further explain the information on lines 230-232 and insert references;
In the discussion of lines 248-261, the drying time is discussed. However, the authors mention drying techniques that are more complex and consequently more expensive than those used in the present work. The discussion could be enriched by discussing such information as energy and equipment cost (this recent work may help DOI: 10.3390/molecules27206876 Molecules 2022, 27(20), 6876);
the quality of Figure 3 is very good, I would like to congratulate the authors for having added it to the work;
the standard deviation must be added in Table 2;
the discussion in Table 2 seems simplistic. For example, what is the interpretation for TPCwhite to be 15, 12 and 14 (mg/g GAE) for temperatures of 60, 70 and 80ºC, respectively?
General: the discussion of the manuscript should be strongly improved. In the current version, the data are discussed only in a comparative way with the literature, not bringing interpretations and estimates for the data. Also, the writing needs to be improved as the text is quite redundant. The conclusion should be improved by bringing the main findings of this study. Please evaluate the obtained results more critically.
After the requested corrections, I will be happy to recommend the manuscript for publication.
Author Response
Thank you very much for your careful reading and your efforts.

Reviewer 2 Report
Nowadays people have been using edible flowers in cooking. Here the authors used some species that had different colors and dried using different temperatures, then checked the bioactive compounds like Total Phenolic Content, Anthocyanin content and Antioxidant activity.
The manuscript needs an English revision
Abstract – need to be written to valorize the results obtained.
Introduction ok
Methods
Line 170 - 2.5 ml change to mL
Line 194 - p<0.05 should in capital letter and italic
Results and discussion
I understand that the journal allows results and discussion together. If the authors prefer to keep both together, then it is important to work with the results, raise hypothesis, comparing the results and then make the discussion. The way that was written is a little confused.
3.1 Developed the results writing, exploring the comparation between all the samples. This topic is confusing, much information together without explain and connecting.
Legend figure 1. What it x axis and y-axis. Legend needs to have more information
What is the meaning of w.b.?
Figure 2 it is difficult to analyzed yellow, orange, and red in this graph. In special when the lines overlap, my suggestion it to change.
My suggestion in topic 3.1 developed better the results, raise hypothesis, and then make discussion. It is confusing the way that is written. The results are good and important for field.
3.2 explore better the results obtained, which is the rational for them? Developed the results and make connections and comparation between the different colors
Explore figure 3 in the text, raise hypothesis and developed.
Improve figure 3 legend, please add a scale bar in the figure 3, it is important for the comparation.
Improve quality from figure 4 and keep the same Y-axis scale. Improve legend from figure 4.
3.3 3.2 explore better the results obtained, which is the rational for them? Developed the results and make connections and comparation between them.
It is hard to see figures 5 and 6, quality is not good. Keep the same scale for y-axis. Improve legends.
Conclusions ok

Author Response

(The authors gave the same response as above.)

Reviewer 3 Report
The manuscript entitled “Investigation of Color and Bioactive Compounds of Different Colors from Pansy (Viola x wittrockiana Gams.) Dried in Hot Air Dryer” it is considered well organized for the journal format; I recommend just a few small changes:
L 130: Recommend to remove this subsection, because it is not a necessary information. The dryer could be mentioned in the 2.3. subsection.
“2.2. Cabinet Convective Dryer
Drying tests were carried out in Dalle brand LT-27 model convective dryer with horizontal air flow cabinet. The drying air temperature can be adjusted digitally between 30-90°C. The rustproof steel device manufactured in Turkey weights 14 kg. There are 12 trays of 390x283 mm in size inside the dryer with dimensions of 462x402x447 mm.”
L197: Also recommend to remove this sentence.
“In the results, the effects of different drying air temperatures and pansy flower colors on drying time, color parameters and bioactive compounds were investigated comparatively.”
Another recommendation is to check the quality of all Figures, because some are blurry.
Figure 2: Please check if the font color is set to black.
Figure 4, 5 and 6: Y and X axis are missing. Please add.
Author Response

(The authors gave the same response as above.)

Round 2
Reviewer 1 Report
After carefully reading the manuscript I recommend it for publication.
Author Response
(x) English language and style are fine/minor spell check required
* The English language and spell check were made again, a few corrections were made in the text.

Reviewer 2 Report
The authors made de modifications that we have sugeested. The manuscript improved. It can be accepted with these modifications.
Author Response
(x) English language and style are fine/minor spell check required
* The English language and spell check made again, a few corrections made in the text.
